# Diverse Pose Lip-Reading Framework

**Naheed Akhter [1], Mushtaq Ali [1], Lal Hussain [2,3], Mohsin Shah [4], Toqeer Mahmood [5] and Amjad Ali [6] and Ala Al-Fuqaha [6,*]**

[1] Department of Computer Science & Information Technology, Hazara University Mansehra, Mansehra 21120, Pakistan

[2] Department of Computer Science & IT, Neelum Campus, The University of Azad Jammu and Kashmir, Athmuqam 13230, Pakistan

[3] Department of Computer Science & IT, King Abdullah Campus, The University of Azad Jammu and Kashmir, Muzaffarabad 13100, Pakistan

[4] Department of Telecommunication, Hazara University Mansehra, Mansehra 21120, Pakistan

[5] Department of Computer Science, National Textile University, Faisalabad 37610, Pakistan

[6] Information and Computing Technology (ICT) Division, College of Science and Engineering (CSE), Hamad Bin Khalifa University (HBKU), Doha 34110, Qatar

* Correspondence: aalfuqaha@hbku.edu.qa

**Abstract:** Lip-reading is a technique to understand speech by observing a speaker's lips movement. It has numerous applications; for example, it is helpful for hearing impaired persons and understanding the speech in noisy environments. Most of the previous works of lips-reading focused on frontal and near frontal face lip-reading and some of them targeted multiple poses in high quality videos. However, their results are not satisfactory on low quality videos containing multiple poses. In this research work, a lip-reading framework is proposed for improving the recognition rate in low quality videos. In this work, a Multiple Pose (MP) dataset of low quality videos containing multiple extreme poses is built. The proposed framework decomposes the input video into frames and enhances them by applying the Contrast Limited Adaptive Histogram Equalization (CLAHE) method. Next, faces are detected from enhanced frames and frontalized the multiple poses using the face frontalization Generative Adversarial Network (FF-GAN). After face frontalization, the mouth region is extracted. The extracted mouth region in the whole video and its respective sentences are then provided to the ResNet during the training process. The proposed framework achieved a sentence prediction accuracy of 90% on a testing dataset containing 100 silent low-quality videos with multiple poses that are better as compared to state-of-the-art methods.

**Keywords:** lip reading; machine learning; face frontalization; generative adversarial network

## 1. Introduction

Lip-reading is a skill to comprehend spoken words from sequences of lips movements. Human speech is bimodal: one is acoustic and other one is visual. The lip-reading affects the actual speech perception, and it is elaborated in [1]. Acoustic speech is closely related to the phoneme where phoneme is the smallest unit of the sound which distinguishes one word from other, for example, the phoneme *k* against words cat, kit, scat, skit and phoneme *p* against words pin, spin and tip. The visual unit is related to the viseme where viseme is a phoneme identical in appearance on lips. Visemes affect the acoustic/phoneme perception, and it alone can be used to recognize speech as shown in Figure 1.

In the early ages, human lip reader experts comprehend the visual speech. There were professional lip-readers who used to interpret many insights from the historic silent movies. In crime scene investigations, the professional lip-readers would interpret conversation between offenders for clear evidence. However, there were problems with the human lip-readers like only experts could interpret the visual speech and the cost of interpretation was high. In addition, they were not reliable as there were chances of biased reading.

Moreover, the human lip-reader required more time to understand visual speech because different people have different speaking styles.

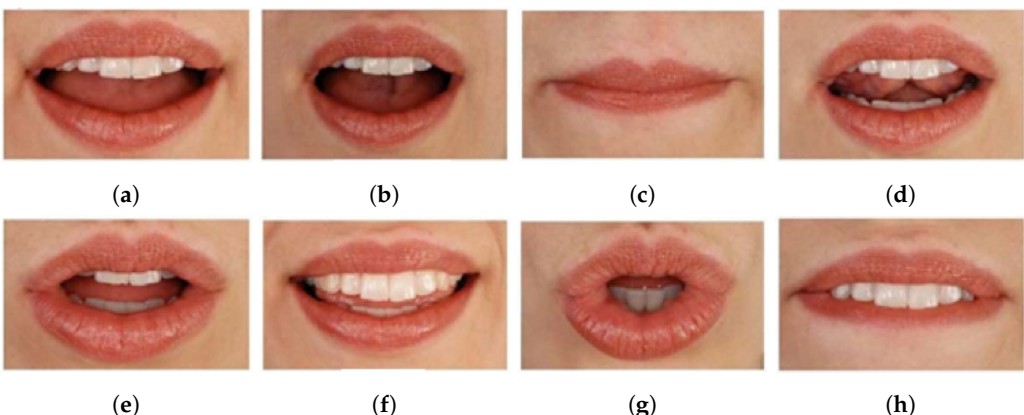

**Figure 1.** Visemes of phonemes: (**a**) ae, (**b**) ar, (**c**) b, (**d**) d, (**e**) e, (**f**) ee, (**g**) er, (**h**) f.

Traditional lip-reading systems work in two steps: feature extraction and classification. In the first step, pixels from the mouth region of interest (ROI) are extracted as visual information. Then, discrete cosine transform (DCT) [2,3], discrete wavelet transform (DWT) [3] and principal component analysis (PCA) [3,4] are used to extract abstract images features. The advance geometric features obtained in this way possess the characteristics of low dimensionality and high robustness. In the second step, the extracted features are fed into the classifiers such as support vector machine (SVM) [5] and hidden Markov model (HMM) [6].

At present, machine learning has made significant progress in the field of computer vision (image representation, target detection, human behavior recognition and video recognition). Therefore, it is an inevitable trend of scientific research to shift the direction of automatic lip-reading technology from the traditional manual feature extraction classification model to the end-to-end machine learning architecture.

Machines learning based lip-reading (MLR) methods extracts features from ROI abstract images which are more discriminative than the features extracted with DCT, DWT and PCA. These features are then fed into a classifier. Most of the current MLR methods performed lip-reading in frontal-view videos. However, relatively lesser works focused on multiple-view lip reading. Multiple-view MLR performed well in high quality videos, but they performed poorly in low quality videos because MLR requires high resolution videos for training. Moreover, in multiple-view MLR, the mouth region is often not clearly visible due to the view angle which significantly degrades the performance of these methods. This issue has been addressed by some methods. However, such methods failed to achieve high accuracy in extreme view angles such as 90° and beyond. How to achieve high accuracy in machine learning based lip reading for low-quality extreme view angles is the research question our method aims to answer.

Our work focuses on lip-reading in multiple poses. In this research work, a new dataset containing low as well as high quality videos of multiple poses is created. Secondly, a speaker dependent lip-reading method has been proposed which can recognize sentences from low as well as high quality videos with multiple poses of a single person. The multiple poses are frontalized using a face frontalization method (FF-GAN) that has not been used in the lip-reading field till now. For improving the quality of video, the proposed lip-reading method uses the CLAHE method. Next, the FF-GAN is used to make frontal faces from multiple poses. The mouth area (lip frame) from frontal face frames is then extracted and sent to the ResNet Deep Neural Network for training.

The main contribution of this work are as follows:

1. Achieving high accuracy of lip reading in low quality videos where the mouth region is not clearly visible;

2. The use of FF-GAN to predict lip reading in extreme pose angles from 90° to the profile.

The rest of the paper is organized as follows: Section 2 presents a comprehensive literature review of the field. The proposed lip-reading framework is presented in Section 3. In Section 4, a deep neural network for lip-reading is described. Experimental results and discussion are presented in Section 5. Finally, the conclusions are drawn in Section 6.

## 2. Literature Review

The current works in lip-reading are based on frontal views/poses of face. The available frontal view lip-readings are speaker dependent [7] as well as speaker independent [8–16]. The speaker dependent systems are trained by individuals who use the system. On the other hand, the speaker independent systems are trained to respond to a word regardless of who speaks. Apart from these works, other works performed lip-reading for isolated speech recognition where a short pause is needed between words and for continuous speech recognition where no such pauses are needed. In [7,11,12,17], the hidden Markov model (HMM) has been used and to adopt certain techniques to make a speaker independent lip-reading system and also tried to improve recognition rate in the continuous visual speech. These works performed well for isolated word recognition and speaker-dependent frontal view lips-reading but did not achieve high results in continuous visual speech. In addition, it has achieved a lower accuracy rate of a speaker independent lip-reading system.

Lip-reading based on deep neural networks (DNN) and Recurrent Neural Networks (RNN) are introduced in [8–10,13,14,18–20]. These approaches performed well on isolated word recognition, but their performance was low over continuous speech. The continuous visual speech for a lip-reading system is problematic due to extensive co-articulation between words as a result homophemes (Homovisemes) produced [7–9,14]. It has been observed that the Convolutional Neural Networks model was unable to classify more accurately on some datasets due to the insufficient data and low-quality videos [8,14]. CNN needs large datasets with high definition videos.

Lip-reading was carried out in different pose angles that are known as multiple view lip-reading. Multiple views were the focus of attention in high quality videos but relatively less works appeared [13,21,22] due to its poor performance in low quality videos. These studies performed sentence recognition in multiple views. These studies performed well on high quality videos and show better results but did not perform well on low quality videos.

Machine based lip-reading is affected by certain factors such as video resolution, multiple people speaking at the same time and their poses. The most important factor for the MLR system is that it requires high-resolution videos. Another factor is multiple people talking at the same time which creates confusion as who is speaking which word and it causes misrecognition of words. This problem is addressed in [8,13] by implementing a synchronization network (SyncNet). The SyncNet learned joint embedding between audio/sound and lip-movements to detect the active speaker.

The pose is another main factor where different poses affect the MLR system. Most of the former studies presented frontal pose lip-reading [7,8,10–14,17,18,23,24] due to the unavailability of data but few works demonstrated lip reading on multiple views [13,21,22] and showed good results on high quality videos. Compared to the frontal view lip-reading, the performance of multiple-view lip-reading was lower because the complete mouth region was often not clearly visible in most cases. However, these studies only work well for high quality videos. It is not essential that high-quality cameras are used in each domain because mobile phone camera, CCTV cameras and webcam recorders generate low quality videos with diverse poses. A face characterization approach exploiting dynamic appearance and time-dependent local features during speech utterance in the Internet-of-Things (IoT) environment is proposed in [25].

The Lombard effect has been introduced in [23]. It is speaking style with increasing speech in noise which is understandable to listener to make robust lip-reading systems. However, it is used more specifically in noisy conditions. Few of the studies made their own

speaker dependent visemes to reduce insertion error in the continuous speech and made the system speaker independent [11,17]. They reduced the insertion error (non-spoken words recognized) to some extent. However, it needs more improvement in speaker independent lip-reading.

A few works used Phoneme and Viseme HMM (Hidden Markov Model) classifiers to improve recognition of visual speech [7,10]. A phoneme classifier performs better than the viseme classifier over speaker dependent. The viseme based classifiers has a lower number of visemes which creates confusion with phonemes and reduces discriminative power of classification. Furthermore, most of the previous studies used HMM, which was a conventional model. HMM was used for speech recognition in early days. Some recent works used it for visual speech recognition [7,11,12,17]. HMM has a state for each class; however, increasing size of states produces complexity of HMM.

Convolutional Neural Networks based lip-reading models were unable to recognize visual speech on low-quality videos [8,14] due to insufficient data. CNN needs large datasets with high definition videos. Some lip-reading methods [21,22,26] in multiple views (visual speech recognition) in high quality videos have demonstrated good results.

## 3. Proposed Lip-Reading Framework

In this section, the implementation details of the proposed lip-reading framework are presented. A sketch of the proposed framework is illustrated in Figure 2, which is composed of five entities: Video Enhancement in which the video quality is improved, Face Detection in which faces are detected from video frames, Face Frontalization where multi-view faces are frontalized after face detection, Mouth Extraction in which the mouth area is extracted from each incoming frames and finally Deep Neural Network (ResNet DNN) is used to predict sentences.

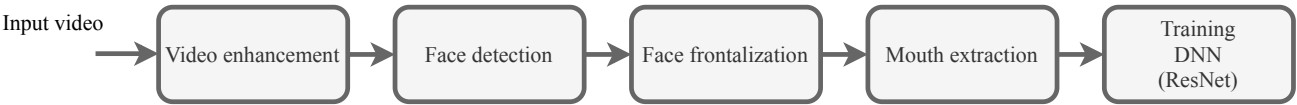

**Figure 2.** Proposed framework for lip-reading.

### 3.1. Video Enhancement

The aim of video enhancement is to improve the quality of the videos frames by reducing noise. It is an essential step of face detection because face detection methods are performed after the enhancement of the video frames. To achieve this aim, the input video is first decomposed into individual frames, and each frame is then enhanced by applying Contrast Limited Adaptive Histogram Equalization (CLAHE) [27]. The CLAHE method is considered because it enhances each small region of the video frame by computing contrast transform function. It sharpens the mouth region of face frames. Other methods such as Adaptive Histogram Equalization AHE [28] and Task Oriented Flow Frontalized [29] can also be used for video enhancement. The AHE method over amplify the noise in the homogenous region of image; however, CLAHE does not amplify noise. On the other hand, the TOFLow method performs better in interpolation but does not show sufficient improvements for blur or noisy image. In order to show the effect of CLAHE method, a low quality video taken with mobile camera is considered as input. The frames of the video are shown in Figure 3. A result of the application of CLAHE method/technique on input frames is shown in Figure 4.

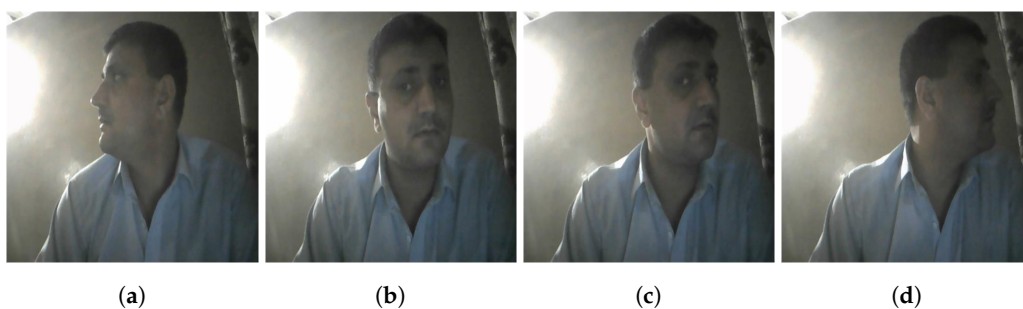

(**a**)          (**b**)          (**c**)          (**d**)

**Figure 3.** (**a**–**d**) Frames of the input low quality video.

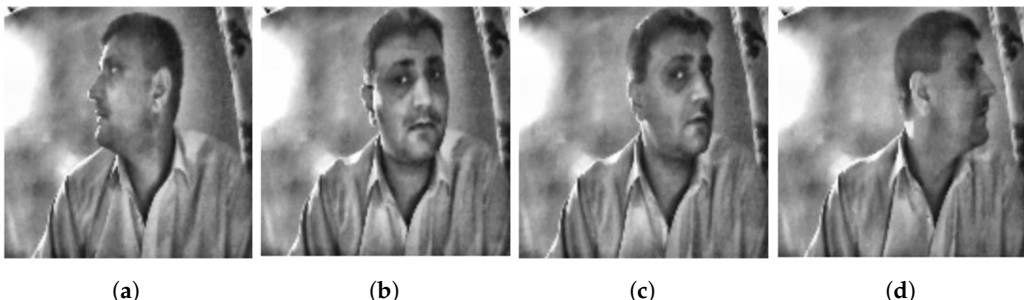

(**a**)          (**b**)          (**c**)          (**d**)

**Figure 4.** (**a**–**d**) Enhanced frames after the application of the CLAHE method.

### 3.2. Face Detection

After video enhancement, face detection is performed in the enhanced video frames. The enhanced video frames come from video enhancement steps and are used for face detection. The face detection is performed by the Histogram of Oriented Gradient (HOG) method, which is widely used method for face detection [14] and object detection. After the application of HOG, the detected image is shown in Figure 5.

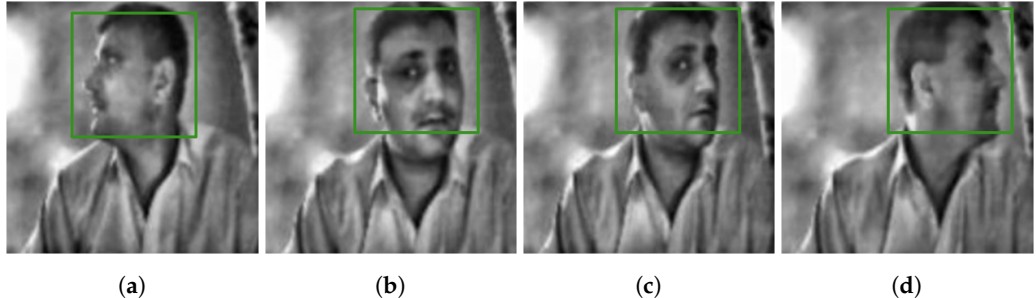

(**a**)          (**b**)          (**c**)          (**d**)

**Figure 5.** (**a**–**d**) Face detection by using HOG Transform.

### 3.3. Face Frontalization (FF)

After face detection, side view faces are frontalized. The Face-Frontlization-Generative Adversial Network (FF-GAN) [30] is used for the face frontalization of multiple poses. The FF-GAN method is used to frontalize more challenging faces (i.e., the face images containing more than half parts of faces are invisible). This method is capable of maintaining global pose accuracy and retains local information of the original image. The FF-GAN focuses on multiple poses of faces to make them frontal and visible. The face frontalization method handles pose variations at extreme poses. Because of its good performance, the face frontalization method is adopted for comprehending lip-reading in all possible poses. The readers are directed to [30] for a detailed mathematical description of the FF-GAN method. The main work of [30] is summarized here.

The general framework of the FF-GAN method is shown in Figure 6, where low frequency information in the form of 3D morphable model (3D-MM) coefficients are provided as a global pose and input non-frontal image is provided to the generator as high frequency information. The discriminator identifies generated faces against real images,

and a recognition engine is used for identity information. The 3D-MM represents a face using Equation (1):

$$p = \{m, \alpha_{id}, \alpha_{exp}, \alpha_{tex}\} \tag{1}$$

$$x^f = G(x, p) \tag{2}$$

where $p$ is 3D-MM coefficients which consist of texture, shape, appearance and projection parameter, and it is used prior to the generator for pose estimation and works as a supporting agent. The Frontalized faces ($x^f$) are obtained using two inputs: input image $x$ and $p$ as represented in Equation (2).

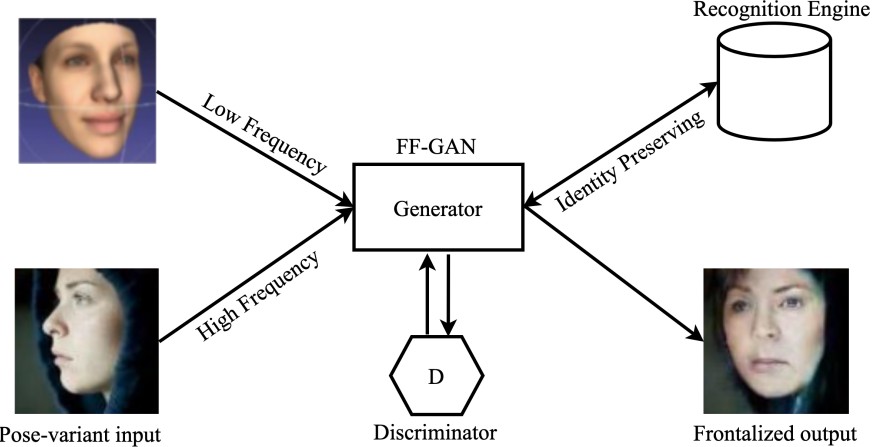

**Figure 6.** General framework of the FF-GAN method [30].

The GAN performed face frontalization by means of an encoder and decoder network and determines reconstruction loss for reconstructing ground truth with a minimal error by Equation (3). The difference between generated (frontal image) and its ground truth to obtain reconstruction loss is computed using Equation (3):

$$LG_{rec} = \|G(x, p) - x^g\|_1 \tag{3}$$

The spatial total variation loss is used for smoothness in the output image. It is determined by Equation (4):

$$LG_{tv} = \frac{1}{|\Omega|} \int_{\Omega} |G(x, p)| du \tag{4}$$

where $|\bigtriangledown G|$ is the image gradient, $u \in R^2$ is the two-dimensional coordinate increment, $\Omega$ is the image region and $|\Omega|$ is the area normalization factor.

As the two halves of a human face share self-similarity, the FF-GAN model introduces symmetry loss to reduce the differences between the two halves of a human face. The symmetry loss is computed by Equation (5):

$$\begin{aligned} LG_{sym} = & \left\|M \odot G(x, p) - M \odot G\left(x_{flip}, p_{flip}\right)\right\|_2 \\ & + \left\|M_{flip} \odot G(x, p) - M_{flip} \odot G\left(x_{flip}, p_{flip}\right)\right\|_2 \end{aligned} \tag{5}$$

where $M$ represents the mask of a generated face (frontal) image containing a visible part of face which is obtained from the 3DMM model. The symbol $\odot$ shows multiplication of two matrices multiplied with a flip version of a generated image and computed the difference after multiplication. The $M_{flip}$ represents a mask of a horizontally flipped version of a generated image. The flipped mask matrices and generated image matrices are multiplied. Next, flipped masks and flipped versions of generated images are multiplied. After multiplication of different matrices, the difference of mask and flipped mask matrices

is computed and is added to obtain symmetry loss. The main purpose of symmetry loss is to obtain similarity between the generated frontal images and its flipped versions.

After finding the losses, frontal image is sent to the discriminator. The discriminator classifies the frontal image as real or generated. The discriminator consists of five convolutional layers and one linear layer which generates the 2D vector. It computes the probability of whether it is a real or generated image.

The self-occlusion in profile faces can destroy the original identity of the faces. The discriminator can only determine whether the generated images are realistic, but it cannot retain the identity of the original images. Therefore, the recognition engine uses cross-entropy loss to classify image $x$ with ground truth identity $y$. The entropy loss is computed using Equation (6) and used to find minimal error for the recognition engine:

$$\min_{\mathbf{c}} \text{Lc} = \sum_j -y_j \log\left(C_j(x)\right) \tag{6}$$

A pre-trained model of FF-GAN is downloaded from the GitHub repository available at https://github.com/scaleway/frontalization, accessed on 20 July 2022. The implementation details are available at the repository online. The training process took three days to complete using the PyTorch framework. The resultant face after the application of FF-GAN is shown in Figure 7.

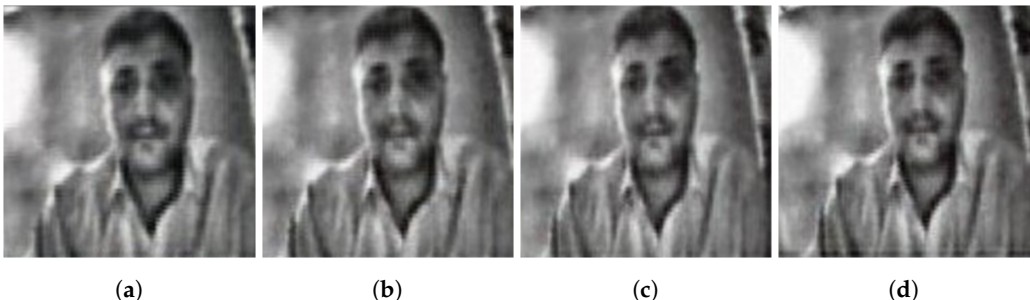

|       |       |       |       |
|:-----:|:-----:|:-----:|:-----:|
| (**a**) | (**b**) | (**c**) | (**d**) |

**Figure 7.** (**a**–**d**) Face frontalization using FF-GAN in six frames of the input video.

### 3.4. Mouth Extraction

In this section, the mouth region is extracted from the frontalized faces. For the mouth extraction, the landmarks are used to localize face regions in the image. The landmarks are markers by which face parts are marked. The shape predictor 68 face landmark pre-trained models have been used on the frontal faces for the landmark localization as shown in Figures 8. The 68 landmark model is developed in [14,29]. These methods had previously been used by [8] for landmark identification of 2D images. The mouth region (lips ROI) has been extracted from the frontal faces in which each frame size contains $224 \times 224$ pixels.

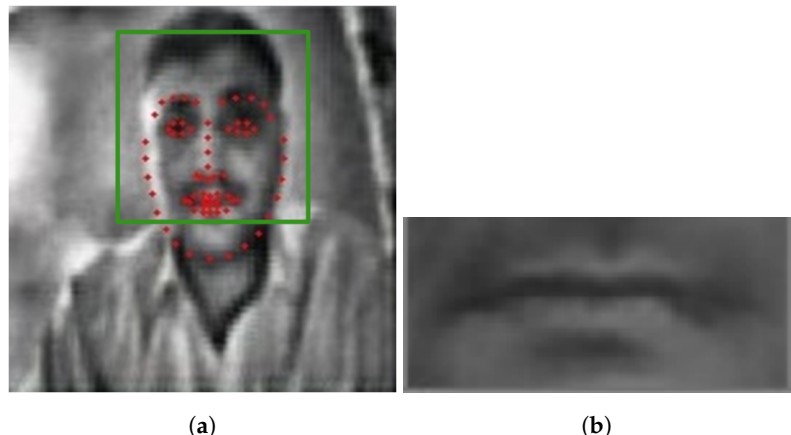

|       |       |
|:-----:|:-----:|
| (**a**) | (**b**) |

**Figure 8.** (**a**) Landmarks identification on frontal faces, (**b**) mouth extraction.

## 4. Deep Neural Network (DNN) for the Lip-Reading

The aim of the deep neural network is to predict spoken sentences from video where the video contains sequences of extracted mouth frames. The input of the network model is a sequence of extracted mouth frames. The proposed framework uses ResNet for recognition of sentences. The ResNet consists of convolutional layers with filter $3 \times 3$ size. Two pooling layers are used in the beginning and end of the network. The input sequence training of images to the network is $224 \times 224$ in size. The Adam optimizer is employed to optimize the model and ResNet perform with two strides. The global average-pooling layer is used at the end after the completion of all residual blocks. The Network architecture is shown in Figure 9. The ResNet solves a vanishing gradient problem which makes model training easier and solves accuracy degradation.

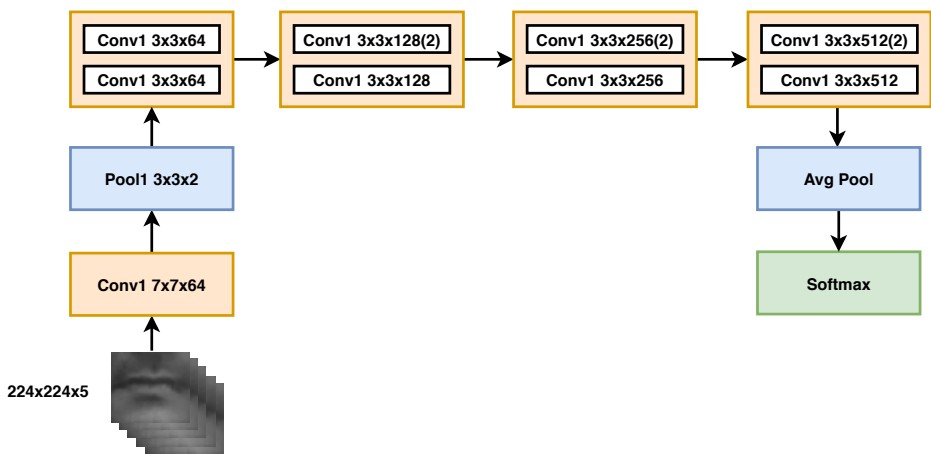

**Figure 9.** DNN (ResNet) architecture for lip-reading.

*Training*

The Deep Neural Network (ResNet) is trained on the sequence of mouth frames extracted from the frontalized faces for visual sentence recognition. The model is trained for 15 epochs and evaluated at a fixed number of steps. The network classification layers are fine-tuned with the Global Average Pooling (GAP) layer. The Dropout and Batch-Norm has been added, which was followed by the Dense and Soft-Max layer. The model has been optimized by the Adam optimizer.

## 5. Experimental Results

In this section, extensive experiments are conducted to evaluate the performance of the proposed lip-reading method. The proposed method is implemented in the PyTorch framework [31]. Experiments are conducted on a server machine with eight 1080Ti GPUs. The proposed method is compared with the state-of-the-art methods on the MP dataset. The MP dataset is divided into three sets: training, validation and test. Different sizes of vocabulary have been defined for all three sets. These three steps are conducted on separate dates as shown in Table 1.

**Table 1.** Division of datasets into training, validation, tests and partition of sentences and vocabulary for three sets.

| Sets | Dates | No. of Uttrances | Vocab |
|---|---|---|---|
| Train | 01/2018–11/2018 | 8880 | 14,708 |
| Val | 01/2019–04/2019 | 4540 | 3401 |
| Test | 05/2019–11/2019 | 3890 | 3401 |
| All | | 17,310 | 21,510 |

### 5.1. Dataset

MP Dataset is our created dataset which consists of 100 speakers in all possible poses from $0°$ to $90°$ view angles. The camera used for capturing videos has a resolution of $720p$, aspect ratio of $16 : 9$ and image size of $1280 \times 720$ pixels. The speaker ages are in the range of 25 to 48 years. Each speaker spoke 300 sentences in the English language. The vocabulary contains 20,000 words. The MP dataset contains 30,000 sentences which are in the form of videos of talking faces with all possible poses. The visual utterances of these sentences have been obtained from CCTV videos, normal mobile phone and webcam, then labeled with corresponding text. These labeled videos are kept as ground truth during training. The whole MP dataset is then decomposed into training and testing datasets containing 17,000 and 3000 sentences taken randomly from the MP dataset, respectively.

Most datasets [7,8,10–14,17,18,23,24] contain frontal faces for lip-reading. Currently, Refs. [13,21,22] contain multiple view datasets and videos recorded in the ideal environment with high definition cameras which consist of frontal, profile and three-quarter viewpoints. On the other hand, the MP dataset contains low quality videos in diverse poses and videos recorded in many types of environments. The proposed lip-reading framework pre-processes videos before pose analysis and visual speech recognition. The detailed summary of lip-reading datasets such as GRID [32], LRS [13], MV-LRS [13] and MP with multiple poses is given in Table 2.

**Table 2.** Vocabulary size with corresponding views in the existing and proposed dataset.

| Datasets | Types | Vocab | No. of Uttrances | Views |
|---|---|---|---|---|
| GRID [32] | Phrases | 51 | 33,000 | $0°$ |
| LRS [13] | Sentences | 807,375 | 118,116 | $0°$–$30°$ |
| MV-LRS [13] | Sentences | 14,960 | 74,564 | $0°$–$90°$ |
| MP dataset | Sentences | 17,500 | 135,442 | $0°$ to profile |

### 5.2. Evaluation Metrics

The proposed model is trained on the MP dataset and validated on independent test sets. Character error rate (CER) has been used to find error in character recognition, word error rate (WER) to find error in words recognition, and the bilingual evaluation understudy (BLEU) metric is used to find precision in sentences. All these evaluation metrics are used for analysis of the proposed framework. Equation (7) is used to find error rate in characters and words:

$$\text{ErrorRate} = \frac{(S + D + I)}{N} \tag{7}$$

where $S$ in Equation (7) represents the number of substitution errors which occur when a user utters one word, and a recognizer recognizes it as another word. $D$ is the number of deletion errors which occur when no word is recognized in response to any spoken word. $I$ is number of insertion errors, which occurs in cases where the word is recognized when none were spoken. $N$ represents the total number of words in references. The Bilingual Evaluation Understudy (BLEU) is modified n-gram precision that evaluates the closeness of the machine translation to human reference translation. In the case of machine learning based lip reading, it is important to compare the predicted sentences with the ground truth sentences. The BLEU metric can be computed using Equation (8):

$$pn = \frac{\sum_{C \in \{ \text{Candidates} \}} \sum_{n-\text{gramec}} \text{Count}_{\text{clip}}(n - \text{gram})}{\sum_{C' \in (\text{Candidates})} \sum_{n-\text{gram}' \in C'} \text{Count}(n - \text{gram}')} \tag{8}$$

The classification accuracy measured by the confusion matrix in Equation (9) is a table that describes the performance of classification model or a classifier for all test sets. In Equation (9), $TP$ represents "true positive", which means that model predicted sentences

which are actually spoken and *TN* represents a "true negative" meaning that a model predicted not any sentence actually when there is silence and just the mouth is open. *N* represents total number of sentences:

$$\text{Accuracy} = \frac{TP + TN}{N} \tag{9}$$

The videos frames are fed forward into the trained ResNet model and the predictions on lip reading are obtained. The obtained predictions are then compared with the ground-truth, and the values of S and D are computed. These values are fed into Equation (7) to compute CER and WER errors. The predicted lip reading sentences are also evaluated using the BLEU metric to check whether the predicted sentences are similar to the ground-truth.

### 5.3. Results

The proposed method achieved good results in multiple poses and extreme view angled faces over other state-of-the-art methods. The proposed model is evaluated on the MP dataset and on the Multi-View Lip Reading Sentences (MV-LRS) [13] dataset. The error rate of all possible poses of the proposed lip-reading framework on the two datasets are given in Table 3. The frontal views of both datasets that show the lowest error rate are also depicted in Table 3.

**Table 3.** Performance measure of two datasets for all possible poses.

| Poses | MP dataset | | | MV-LRS | | |
|---|---|---|---|---|---|---|
| | CER | WER | BLEU | CER | WER | BLEU |
| Frontal | 42.1% | 51.0% | 48.4 | 46.5% | 56.4% | 49.3 |
| 15° | 45.5% | 53.7% | 50.2 | 49% | 56% | 47 |
| 30° | 49.7% | 57% | 45.8 | 50.4% | 59.2% | 46.1 |
| 45° | 46% | 55.6% | 45.8 | 50.4% | 59.2% | 46.1 |
| 60° | 48% | 55% | 46.1 | 50.4% | 59.2% | 46.1 |
| 75° | 53.7% | 60.1% | 46.1 | 53.2% | 62.3% | 45.8 |
| 90° | 53% | 59.3% | 41.4 | 54.4% | 62.8% | 42.5 |

Table 4 demonstrates sentences accurately predicted by the proposed framework. The MP Dataset gives significant performance over these sentences.

**Table 4.** The proposed method accurately recognized these sentences.

| S.No | Sentences |
|---|---|
| 1 | YOU JUST HAVE TO FIND ANOTHER TERM AND LOOK THAT UP |
| 2 | I DO LIKE MAGIC |
| 3 | WE REALLY DON'T WALK ANYMORE |
| 4 | AT SOME POINT I'M GOING TO GET OUT |
| 5 | WHEN I GET OUT OF THIS AM I GOING TO BE REJECTED |
| 6 | WE'D LOVE TO HELP |

### 5.4. Comparison

The performance, in terms of accuracy, of multiple poses has been compared on the MP dataset with existing methods [20,21,26] as illustrated in Table 5. All the methods, except the proposed DNN (ResNet) method, show a decreasing trend in accuracy with the increasing view angle. The accuracy of the method [20] is the lowest among all the methods

and decreases rapidly with increasing view angle. The method in [21] shows a relatively better accuracy and decreases gradually with increasing view angle. A high accuracy of 90.1% is shown by the method in [26] for frontal faces. The effect of view angle on the accuracy of this method is relatively less compared to the method in [21]. Compared to all the mentioned methods, the accuracy of the proposed framework is superior in the frontal and side view angles. It can be noted that the accuracy of all the other methods decreases with the increasing view angle, but the accuracy of the proposed framework remains stable and does not vary much. The reason is the fact that our proposed framework is equipped with the CLAHE method and FF-Gan. Due to the applications of these two methods, the proposed framework is getting frontalized face images with better contrast, which in turn enables it to localize the mouth portion on the face images with accuracy better than the prior methods. The sentence prediction accuracy of the proposed DNN (ResNet) is comparatively better because of training on an accurate sequence of mouth frames extracted from each speaker dependent video. Furthermore, all the methods in [20,21,26] can process either frontal faces video frames, videos containing isolated words and videos that have been recorded in a specific environment or high quality videos. Our proposed framework improved the accuracy of sentence recognition from the speaker dependent lip movements. However, its sentence recognition accuracy is not satisfactory for unseen speaker lip movements. Furthermore, the FF-Gan results are not up-to-the-mark for faces with hair-like mustaches and beards, which in turn affect the recognition accuracy of our proposed framework for spoken words of people with mustaches and/or beards.

**Table 5.** Performance measure over multiple poses by using different methods.

| Methods | Frontal | 15° | 30° | 45° | 60° | 75° | 90° |
|---------|---------|-----|-----|-----|-----|-----|-----|
| Y. Lan [21] | 70.01% | 69.04% | 65.50% | 62.98% | 61.73% | 60.09% | 58.85% |
| MV-WAS [26] | 90.1% | 89.07% | 87.8% | 85.0% | 82.0% | 80.10% | 78.9% |
| DNN and LSTM [20] | 60.00% | 55.05% | 49.01% | 35.00% | 32.00% | 30.00% | 28.06% |
| DNN(ResNet) | 92.05% | 92.05% | 91.08% | 91.08% | 91.00% | 90.08% | 90.00% |

## 6. Conclusions

A framework for diverse pose lip-reading in low quality videos has been presented. Additionally, a dataset that contains high and low-quality videos in multiple poses has been created. Our proposed framework used the FF-GAN method for multiple poses frontalization. The recognition of sentences has been performed by ResNet. The lip-reading performance has been measured for multiple poses that has shown better results and outperformed state-of-the-art methods.The diverse pose lip-reading is speaker dependent which needs more robust speaker independent lip-reading either in multiple poses and frontal poses. It has been observed that different pose angles demonstrate different accuracy levels, but the frontal pose always occupies the top level of accuracy among all possible poses. Its needs to make systems that maintain the same accuracy level for all possible poses. A part of that facial hair i.e., mustaches and beards, becomes the obstacles in the lip-reading framework improvement when using the frontalization method. Furthermore, no literature has been found on extreme profile views in lip-reading. In the future, we intend to work on extreme profile views based lip-reading.

**Author Contributions:** Data curation, N.A., M.A., L.H., M.S., T.M. and A.A.; Investigation, N.A., M.A. and L.H.; Methodology, N.A., M.A., L.H., M.S. and A.A.-F.; Project administration, A.A.; Supervision, T.M., A.A. and A.A.-F.; Writing—review & editing, N.A., M.A., L.H. and A.A.-F. All authors have read and agreed to the published version of the manuscript.

**Funding:** This research received no external funding.

**Institutional Review Board Statement:** Not applicable.

**Informed Consent Statement:** Informed consent was obtained from all subjects involved in the study.

**Data Availability Statement:** Data available on request.

**Acknowledgments:** We are thankful to Qatar National Library (QNL) for supporting the publication charges of this paper.

**Conflicts of Interest:** The authors declare no conflict of interest.

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
