# Peer review of "Diverse Pose Lip-Reading Framework"

_applsci, doi:10.3390/app12199532_

Round 1

Reviewer 1 Report

Authors claimed that the accuracy of the proposed method is superior in the frontal and side view angles. But more details are necessary in order to see benefits of the proposed methodology. Validation of the DNN (ResNet) should be better explained. Limitations of the study are not mentioned.

Author Response

REVIEWER 1 COMMENTS

Comment 1:The authors claimed that the accuracy of the proposed method is superior in the frontal and side view angles. But more details are necessary in order to see the benefits of the proposed methodology. Validation of the DNN (ResNet) should be better explained. Limitations of the study are not mentioned.Response to comment 1:Thank you for your valuable comments. In the revised manuscript in the 5.4 Comparison section, these three comments have been addressed. The details of the proposed methodology benefits are highlighted in green in lines 278 – 281. The validation of the DNN is highlighted in blue in lines 282 – 284 and the limitation of the proposed work is highlighted in red in 286 – 291.

Reviewer 2 Report

Dear Authors

Very interesting topic in the field of image processing! The Authors made an appropriated work to study the diverse pose lip-reading framework which is highly appreciated. However, it would be great if the Authors address the Reviewer´s concerns in the revised version as below:

1.      The topic is novel, but please stress the novelty of the work in a more precise manner, very brief and straight and its application and contribution to the state of the art.

2.      It would be great if you use the passive tense instead of active tense, I meant the sentences must not start with “WE”, please correct this issue.

3.      Figure 3 is Vague! First, please reproduce it with a better quality, enlarge it and add more details on it.

4.      Please add the algorithms or any routines you developed to process and obtain the results. Therefore, it permits others to repeat what you have done.

5.      Can you add the relative error amongst your results and others? Also an equation to obtain the error is needed.

6.      Can you give some information regarding the camera which took the video of your experiment? In addition, can you discuss the camera parameters and their influence on the results? Maybe it will raise doubts in the Readers’ mind.  

Very Best

The ReviewerDear Authors

Very interesting topic in the field of image processing! The Authors made an appropriated work to study the diverse pose lip-reading framework which is highly appreciated. However, it would be great if the Authors address the Reviewer´s concerns in the revised version as below:

1.      The topic is novel, but please stress the novelty of the work in a more precise manner, very brief and straight and its application and contribution to the state of the art.

2.      It would be great if you use the passive tense instead of active tense, I meant the sentences must not start with “WE”, please correct this issue.

3.      Figure 3 is Vague! First, please reproduce it with a better quality, enlarge it and add more details on it.

4.      Please add the algorithms or any routines you developed to process and obtain the results. Therefore, it permits others to repeat what you have done.

5.      Can you add the relative error amongst your results and others? Also an equation to obtain the error is needed.

6.      Can you give some information regarding the camera which took the video of your experiment? In addition, can you discuss the camera parameters and their influence on the results? Maybe it will raise doubts in the Readers’ mind.  

 Very Best

The Reviewer 

Author Response

REVIEWER 2 COMMENTS

Comment 1:

The topic is novel, but please stress the novelty of the work in a more precise manner, very brief and straight and its application and contribution to the state of the art.

Response to comment 1:

Thank you for your valuable comments. We have added the main contributions of the work in the Introduction section in lines 67 – 71. Furthermore, more details about the contributions and novelty of the work have been added in the comparison section that are highlighted in lines 278 – 284.

Comment 2:It would be great if you use the passive tense instead of active tense, I meant the sentences must not start with “WE”, please correct this issue.Response to comment 2:Thank you for your valuable comment. In the revised manuscript, we have made these changes and have used passive voice instead of the active voice.Comment 3:

Figure 3 is Vague! First, please reproduce it with a better quality, enlarge it and add more details on it.

Response to comment 3:Thank you for your valuable comment. Figure 3 has been reproduced with 300 dpi in the revised manuscript at page 5.Comment 4:

Please add the algorithms or any routines you developed to process and obtain the results. Therefore, it permits others to repeat what you have done.

Response to comment 4:Thank you for your valuable comment. In the revised manuscript Figure 9 is used to process and obtain the results.Comment 5:Can you add the relative error amongst your results and others? Also an equation to obtain the error is needed.Response to comment 5:Thank you for your valuable comment. A comparison of the three types of errors namely CER, WER and BLEU is illustrated in Table 3 at page 10. These errors are computed between our MP dataset and MV-LRS datasets for different view angles.Comment 6:

Can you give some information regarding the camera which took the video of your experiment? In addition, can you discuss the camera parameters and their influence on the results? Maybe it will raise doubts in the Readers’ mind.

Response to comment 6:

Thank you for your valuable comment. We have used the camera of Samsung mid-range mobile for capturing videos for our dataset. Although we have not added details about the specific camera used for experiments, we have added parameters of the camera such as resolution, aspect ratio and frame size in pixels at page 8 lines 240 – 241.

Reviewer 3 Report

The research paper describes a framework to read lip movement from multiple poses and low quality videos. The introduction, literature review, experimental design, results and analysis are well organized. Minor grammar corrections in some sentences and adding missing references to few abbreviations would improve the quality of the paper, further. The research work uses FF-GAN for frontalization, and ResNet for sentence recognition. It focuses on English sentences and shares a newly developed dataset. The reviewer recommends adding a description of what MP means (Multiple Pose?) when it is mentioned the first time. Overall the paper provides insights on an interesting research topic and would be useful in future research which considers occlusion, other languages and more variety of participants. The reviewer recommends an accept with minor changes.

Author Response

REVIEWER 3 COMMENTS

Comment 1:

Minor grammar corrections in some sentences and adding missing references to few abbreviations would improve the quality of the paper, further. The reviewer recommends adding a description of what MP means (Multiple Pose?) when it is mentioned the first time.

Response to comment 1:

Thank you for your valuable comments. In the revised manuscript, minor grammatical mistakes are corrected. The description of MP dataset is added in the Abstract and highlighted where it is used for the first time.

Reviewer 4 Report

The topic of the article is interesting and current. Therefore, I have the following minor and several serious comments about it.

It is unusual and usually inaccurate to claim that an approach has not been used before. I recommend reformulating or clarifying - page 2, lines 61-62: "...frontalized using face frontalization method (FF-GAN) that have not been used in the lip-reading field until now".Check: https://openaccess.thecvf.com/content/ICCV2021W/TradiCV/html/Kang_Robust_Face_Frontalization_for_Visual_Speech_Recognition_ICCVW_2021_paper.html or https://ieeexplore.ieee.org/abstract/document/9674610

The definition of goal or research question is missing in the introduction.

A more detailed explanation of equation 5 is missing in terms of why the given procedure is used. The authors describe what is happening with the data but do not describe why. In addition, the link to GitHub is broken. I recommend including all the information in the article and basic implementation details too.

Transfer learning used for training (line 219) is not quite clearly described; I recommend providing details.

Table 1 should probably be placed as part of Chapter 5, not above it

Line 232 – what means speakers in all possible poses? From behind too? Try to be more specific.

Creating a dataset is very useful research output. Is the dataset available? Instead of a general description of what devices were used, it would probably be more appropriate to state their resolution - since the article is oriented toward lip-reading from a lower-quality image.

Check table 2 – what means 14,428 Types in aeLRS? Which GRID is considered? Use the links to resources.

I am unsure if the information in table 2 and the paragraph above is placed correctly. Although the experiment is already described at this point in the article, this information is more suitable for the parts before it.

In the Evaluation metric chapter, the procedure should be explained (the data source and the procedure for their preparation for entry into the metrics, where is the recognized text obtained, where is it stored, and where are the read phrases/sentences obtained?).

Table 3 – not BELU -> BLEU

Explain the purpose of using the BLEU method. It is more justified when evaluating the quality of a translation, where the exchange of words is a common phenomenon. When recognizing the sequence of words from the image, it is unclear to me whether the order of words can be changed at all...

I accept the metric from equation 7; I am unclear about the goal of equation 8; equation 9 represents a typical accuracy determination.

The article as a whole makes a good impression, and it clearly describes the process of the experiment. However, a serious shortcoming is the missing description of the central part of the experiment.

What differentiates the acquisition of data from low-resolution sources compared to the procedures applied by other authors? Unfortunately, this idea disappears from the article. The authors mention it in the introduction as a difference from other solutions but didn't discuss it finally. As a result, no difference between high and low-resolution work can be observed in the presented framework.

Suppose the difference lies in the calibration and setting of the parameters of individual networks/tools. In that case, it should be stated how, where and by what the inputs and outputs of the high and low-resolution sources differ.

Without a more detailed description of the steps related to the main idea of the article, it cannot be recommended for publication.

Author Response

REVIEWER 4 COMMENTS

Comment 1:It is unusual and usually inaccurate to claim that an approach has not been used before. I recommend reformulating or clarifying - page 2, lines 61-62: "...frontalized using face frontalization method (FF-GAN) that have not been used in the lip-reading field until now"Check: https://openaccess.thecvf.com/content/ICCV2021W/TradiCV/html/Kang_Robust_Face_Frontalization_for_Visual_Speech_Recognition_ICCVW_2021_paper.html Orhttps://ieeexplore.ieee.org/abstract/document/9674610Response to comment 1:Thank you for your valuable comments. We agree with the reviewer’s comment about the use of face frontalization in the field of automatic lips reading. However, the methods mentioned by the reviewer use different approaches of face frontalization. The method in

https://openaccess.thecvf.com/content/ICCV2021W/TradiCV/html/Kang_Robust_Face_Frontalization_for_Visual_Speech_Recognition_ICCVW_2021_paper.html

proposed a robust face frontalization method and the method in https://ieeexplore.ieee.org/abstract/document/9674610

proposed 3D face alignment to obtain spatial depth information, and color attributes to modulate depth information to add the missing information due to pose changes. This method works on only two poses 1) straight and 2)  degree angle. Our method proposes face frontalization using the Generative Adversarial Networks (FF-GAN) that have superior performance in fronatlizing extreme poses beyond  compared to the methods as mentioned by the reviewer.

Comment 2:The definition of goal or research question is missing in the introduction.Response to comment 2:

Thank you for your valuable comment. In the revised manuscript, the research question and goal of our manuscript are provided at the end of the Introduction section and highlighted. Lines 56 – 58.

Comment 3:A more detailed explanation of equation 5 is missing in terms of why the given procedure is used. The authors describe what is happening with the data but do not describe why. Response to comment 3:

Thank you for your valuable comment. More details of Eq. 5 (above Eq. 5) are given that explain the use of symmetry loss in our work. Lines 194-195.

Comment 4:The link to GitHub is broken. I recommend including all the information in the article and basic implementation details too.Response to comment 4:Thank you for your valuable comment. In the revised manuscript, the GitHub link is restored and is working properly. Line 201.Comment 5:Transfer learning used for training (line 219) is not quite clearly described; I recommend providing details.Response to comment 5:Thank you for your valuable comment. The phrase ‘Transfer learning’ is mistakenly added. In the revised manuscript, the correction is made. Line 225 – 226.Comment 6:Table 1 should probably be placed as part of Chapter 5, not above it.Response to comment 6:Thank you for your valuable comment. In the revised manuscript, Table 1 is placed at the right location in section 5 Experimental Results. Comment 7:Line 232 – what means speakers in all possible poses? From behind too? Try to be more specific.Response to comment 7:Thank you for your valuable comment. In the revised manuscript, correction is made and view angles are added and highlighted in lines 238 – 239. We have done some experiments on view angle more than  degree, however, it will be added in our future work. The following figures illustrates face frontalization from view angle more than  degree.

    Comment 8:

Creating a dataset is very useful research output. Is the dataset available? Instead of a general description of what devices were used, it would probably be more appropriate to state their resolution - since the article is oriented toward lip-reading from a lower-quality image.

Response to comment 8:

Thank you for your valuable comments. We are capturing more videos to add to the MP dataset so as to make a large dataset. We plan to make it public in the near future after more low quality diverse pose videos are added to the dataset. In the revised manuscript, camera details such as resolution, aspect ratio and frame size in pixels at page 8 lines 240 – 241.

Comment 9:

Check table 2 – what means 14,428 Types in aeLRS? Which GRID is considered? Use the links to resources.

Response to comment 9:

Thank you for your valuable comments. The values in the second column of Table 2 were by mistake. In the revised manuscript, correction is made and resource links of datasets are added and highlighted. Line 255.

Comment 10:

I am unsure if the information in table 2 and the paragraph above is placed correctly. Although the experiment is already described at this point in the article, this information is more suitable for the parts before it.

Response to comment 10:Thank you for your valuable comment. In the revised manuscript, details are added about Table (line 255). The paragraph above Table gives a general overview of the different datasets and the MP dataset.Comment 11:

In the Evaluation metric chapter, the procedure should be explained (the data source and the procedure for their preparation for entry into the metrics, where is the recognized text obtained, where is it stored, and where are the read phrases/sentences obtained?).

Response to comment 11:Thank you for your valuable comment. In the revised manuscript, details about how to compute CER, WER and BLEU are provided in lines 258 – 262.Comment 12:Table 3 – not BELU -> BLEUResponse to comment 12:Thank you for your valuable comment. In the revised manuscript, correction is made in Table 3 at page 10. Comment 13:Explain the purpose of using the BLEU method. It is more justified when evaluating the quality of a translation, where the exchange of words is a common phenomenon. When recognizing the sequence of words from the image, it is unclear to me whether the order of words can be changed at all...Response to comment 13:Thank you for your valuable comments. In the revised manuscript, the purpose of using the BLEU n-gram metric is explained and highlighted (After Eq. 7).Comment 14:The article as a whole makes a good impression, and it clearly describes the process of the experiment. However, a serious shortcoming is the missing description of the central part of the experiment.Response to comment 14:Thank you for your valuable comments. The description about the central part of the experiments are provided in section 5.3 Results. Firstly, the MP is dataset is compared in terms of different errors with the MV-LRS dataset on different view angles. Secondly, the proposed method is used to predict certain lip reading sentences and thirdly the accuracy of the proposed method is compared with state-of-the-art methods.Comment 15:

What differentiates the acquisition of data from low-resolution sources compared to the procedures applied by other authors? Unfortunately, this idea disappears from the article. The authors mention it in the introduction as a difference from other solutions but didn't discuss it finally. As a result, no difference between high and low-resolution work can be observed in the presented framework. Suppose the difference lies in the calibration and setting of the parameters of individual networks/tools. In that case, it should be stated how, where and by what the inputs and outputs of the high and low-resolution sources differ.

 Response to comment 15:Thank you for your valuable comments. We have created the MP dataset that contains low resolution videos in diverse poses. In the revised manuscript, we have added the details of the camera used for capturing low resolution videos (page 8 lines 240 – 241).  Table 3 and Table 5 do illustrates the performance difference between low resolution videos and high resolution videos. Table 3 compares the low resolution MP dataset with the high resolution MV-LRS dataset on different view angles and shows that the performance on MP dataset is superior. Similarly, in Table 5 the proposed method is compared with other state-of-the-art methods on low resolution videos.

Round 2

Reviewer 1 Report

Authors responded properly on my comments.

Reviewer 2 Report

Dear Authors

Good Improvement! The Reviewer proposed an acceptance at this stage.

Very best regards

The Reviewer 

Reviewer 4 Report

I appreciate the detailed inclusion of comments and thank the authors for explaining unclear parts of the article. I believe that the new version of the article is in a high-quality, comprehensible, relatively fluent, and easy-to-read form.

From my point of view, I consider the review complete and recommend the article for publication.